# Role of Nutrition and Exercise Programs in Reducing Blood Pressure: A Systematic Review

**DOI:** 10.3390/jcm8091393

**Published:** 2019-09-05

**Authors:** Roman Jurik, Petr Stastny

**Affiliations:** Department of Sport, Faculty of Physical Education and Sport, Charles University, 162 52 Prague, Czech Republic

**Keywords:** resistance training, hypertension, arterial pressure, disease prevention, caffeine

## Abstract

The combined effect of diet and strength training (ST) on blood pressure (BP) seems to be very important for the treatment of prehypertension and hypertension (HT). Therefore, the aim of this study was to determine whether ST alone or combined with nutrition or supplementation has an impact on the arterial pressure reduction in normotensive and hypertensive populations. A systematic computerized literature search was performed according to the PRISMA guidelines using PubMed, Scopus and Google Scholar; only English language studies published from 1999 until 2018 were included. This systematic search identified the results of 303 individuals from nine studies. The ST program alone had a similar effectiveness as the nutrition program (NP) alone; however, their combination did not result in increased effectiveness in terms of a high BP reduction. The consumption of L-citrulline had a similar effect as ST on lowering BP; on the other hand, caffeine led to an increase in BP during the ST session. Our data suggest that a combination of ST 2–3 times a week at moderate intensity and a NP seems to be equally effective in terms of lowering BP (systolic and diastolic) as ST and NP alone.

## 1. Introduction

The nonpharmacological approach to hypertension (HT) reduction is based on lifestyle changes using nutritional and exercise strategies; different training interventions [1,2,3,4,5,6,7] or nutritional plans [8,9,10] have been shown to decrease arterial pressure values. Previous reviews have summarized the approaches of aerobic training [11,12,13], anaerobic training [14,15,16,17] and nutrition [18,19,20,21], and those strategies were effective for HT reduction with expected decreases of 5 mmHg in systolic blood pressure (SBP) and 3 mmHg in diastolic pressure (DBP) [22] after three months. However, nonpharmacological strategies do not contraindicate each other, and their combination has been shown to be effective for other health improvements, such as bodyweight reduction [10,23,24,25,26,27]. Therefore, there is a question as to what kind of intervention or their combination has an improved effect on the HT decrease.

One of the well-documented interventions that has been shown to reduce arterial pressure is strength training (ST), which has already been reviewed to set optimal training loads, such as the number of sets [28,29,30,31], repetitions [32,33,34,35] and rest intervals during training sessions [32,33,34,36,37,38]. One of the ST effects is eliciting high muscle protein degradation followed by protein synthesis, which increases basal metabolism and is therefore usually accompanied by changes in nutritional requirements. In clinical practice, it is typical that ST is prescribed along with a low carb diet [39,40,41], specific protein intake [42,43,44,45,46,47,48,49] or another strategy that might support and magnify the arterial pressure decrease. However, there is currently no recommendation regarding whether the ST program should be accompanied by specific nutritional support that could result in a greater effect of HT reduction.

Numerous studies have shown that different nutritional programs (NP) and strategies might lead to nonpharmacological decreases in arterial pressure [50,51,52,53,54,55], while many strategies place a high demand on patients to make changes in their eating habits. One way to easily change food intake might be accomplished by using food supplements, which have a synergistic effect with the application of ST. Some food supplements have been shown to decrease arterial pressure when used with aerobic training [56,57,58,59] or alone. On the other hand, some food supplements might have a positive effect alone but when used simultaneously with exercise might cause side effects, such as post-exercise hypotension [60,61,62,63,64,65,66].

In the current literature, there is no current overview of whether ST or nutritional strategies have a stronger effect on HT reduction. Therefore, the aim of this study was to determine whether ST alone or ST combined with nutrition or a supplement has the greatest impact on arterial pressure reduction in normotensive and hypertensive populations. Additionally, this review aimed to summarize what kind of combined ST and NP might be effective for HT reduction without side effects. The main hypothesis of this article is that the biggest effect on HT should be observed when strength training is combined with energy intake restriction. Based on this result, practitioners can establish nonpharmacological treatment for the individuals with increased blood pressure (BP).

## 2. Methods

This systematic review is reported in accordance with recommendations as presented in the Preferred Reporting Items for Systematic Reviews and Meta-Analysis (PRISMA) statement [67]. The protocol for this systematic review was published on PROSPERO under registration number CRD42019130631.

### 2.1. Search Strategy

A systematic computerized literature search was performed using PubMed, and Scopus and included studies published in English from 1966 until November 2018. The search formula included the following terms: Blood pressure AND hypertension OR cardiovascular disease OR hypotension AND resistance training OR strength training OR weightlifting OR bodybuilding OR exercise. The search was limited by article types, species, subjects, language, age, and text availability. A manual search was performed using the identified reviews, reference lists of the selected articles and Google Scholar.

### 2.2. Types of Studies

The review considered cohort studies, analytical cross-sectional studies, randomized control trials, nonrandomized control trials, intervention studies, case-control studies and others that included BP and HR measurements as well as data on NP and ST in all adult populations. The review studies were used for manual searches of their reference list. Dissertations, theses, conference proceedings, conference monographies, and other reviews were not included. Retrospective studies were not included because the area of interest requires performing experiments. The qualitative component also considered the type of ST and NP and methodological designs. All titles and abstracts were screened according to the above-mentioned inclusion criteria after removing the duplicates. Full texts of eligible articles were retrieved and assessed by two reviewers (R. J. and P.S.). Any discrepancies between the two reviewers were managed by a consensus discussion.

### 2.3. Types of Outcomes

The review considered studies that included the following outcome measures: Acute SBP and DBP variability before, during, and after exercise or a nutritional treatment; the HR variability before, during, and after exercise or treatment; and a mean arterial pressure before, during, and after exercise. The exclusion criteria were as follows: Full text was not available in English, the study did not contain an appropriate description of measurement devices and procedures, the study did not include a proper exercise or nutrition task, or the study did not report how raw data were processed.

### 2.4. Data Extraction and Evaluation

Data extraction included aspects of the study population, such as the average age and sex, specific aspects of the NP and ST intervention (sample size, type of exercise performed, presence of supervision, frequency, and, duration of each session, type of diet, type of supplements), outcome measures and results presented, and the values of BP or HR (Appendix A); however, the studies were not rejected if any part were missing. The Physiotherapy Evidence Database (PEDro) scale was used to assess the methodological quality of a study based on general criteria, such as concealed allocation, intention-to-treat analysis, and adequacy of follow-up. These characteristics make the PEDro scale a useful tool to assess methodological quality [68]. Extraction was performed by two reviewers (R. J. and P.S.). The lack of clarity during the extraction process was resolved by the reviewer’s discussion. The PEDro scale based on a Delphi list [69] was used for all articles even if the trials had already been rated by trained evaluators of the PEDro database (http://www.pedro.fhs.usyd.edu.au/).

## 3. Results

### 3.1. Study Characteristics

The systematic literature search through database searching identified 15,302 records. However, after duplicates were removed, 11,558 records were screened based on the title and abstract. The title and abstract screening resulted in 144 records for full-text eligibility. From these records, nine studies satisfied the quality and exclusion criteria and were selected for this systematic review after full-text screening (Figure 1).

Table 1 provides a general description of each study, sample, and intervention characteristics. Only two studies included seniors [70] and [71]. One study included only men [72], and three included a mixed sample of men and women [70,71,73]. Five studies included only women [74,75,76,77,78]. In total, five studies compared the effect of ST with the NP intervention effect on BP [70,73,74,75,76]. Another three studies compared the effect of ST and supplementation intervention on BP [71,72,78,79] and one study analysed the effect of ST and supplementation sesion in cross sectional design [71].

Based on the study results compilation, it can be stated that ST alone has a positive effect on the BP values (Figure 2A,B). However, the effect did not depend on the type of ST (resistance training, bodyweight training or training on vibration platforms, etc. Table 2). An important prerequisite for effective ST is the selection of suitable parameters and methods; otherwise, there is a risk of injury or side effects. All studies used training parameters (Table 2) in accordance with recommendations established by The American College of Sport Medicine (ACSM) and the Canadian Hypertension Education Program, according to Pescatello et al. [80]. However, each study chose different training parameters based on the performance and health state of their subjects.

### 3.2. Strength Training Intervention

In studies by Figueroa et al. [76], Arazi et al. [77], Wong et al. [78] and Astorino et al. [72], a reduction in SBP after ST alone was detected in prehypertensive individuals. Only Astorino et al. [72] observed a negligible improvement in the BP values in normotensive individuals. Figueroa et al. [76] also detected a reduction in DBP after ST. It seems that ST is one of the initiators of post-exercise hypotension in adults, regardless of the type of ST (see in Figure 2A,B, Figure 3 and Appendix A).

Body composition and strength parameters have been improved by ST alone in the study by Figueroa et al. [76]. This confirmed the assumption that ST has a generally positive effect on health status because it might reduce body fat and increase muscle mass and other conditioning values. Strength abilities were improved in participants of studies combining ST and nutrition, and these findings have been reported by Sales et al. [75], Figueroa et al. [76], Moraes et al. [70], and Lee et al. [73]. Strength abilities worsened in the nontraining group (only NP) in the study by Figueroa et al. [76]. In contrast, the Dietary Approaches to Stop Hypertension (DASH) diet group had no negative effect on strength according to Lee et al. [73]. A comparison of the effect of ST alone or ST and NP on absolute and relative strength bring very similar results as those shown by Figueroa et al. [76]. The aerobic parameters were improved in the ST group with NP in the studies by Moraes et al. [70] and Lee et al. [73], and the VO2max was improved in the study by Sales et al. [75].

### 3.3. Nutrition Program

Figure 2A,B show that SBP and DBP decreased after a nutrition program [74,76]. Only a study by Lee et al. [73] reported a negligible increase in SBP after eight weeks of a NP in individuals with prehypertension (PHT) and HT. Moraes et al. [70] showed that a higher intake of dairy products together with combined aerobic and anaerobic training can lead to a slightly larger drop in BP than that associated with a lower intake of dairy products. Moreover, the higher intake of dairy products resulted in a smaller increase in BP (i.e., return of BP to the initial values) after six weeks of nontraining [70]. It appears that the inclusion of more dairy products in the NP together with combined aerobic and anaerobic training has a positive effect on the reductions in the SBP and DBP values.

A study by Villani, Gornall [74], Figueroa et al. [76], and Lee et al. [73] revealed that a training program combined with a NP leads to a significant drop in BP. Moreover, one study showed that ST alone had similar effectiveness as that of the NP [76].

### 3.4. The Effectivity of Food Supplements

Caffeine is a very popular pre-workout stimulant among athletes. Astorino et al. [72] found that caffeine intake immediately before training increases SBP (*p* < 0.05). However, the effect on HR (*p* = 0.16) and DBP (*p* = 0.10) was similar for the caffeine and placebo. Values of a HR and BP were significantly higher in men with PHT than in normotensive men (*p* < 0.05).

A study by Arazi et al. [77] performed on middle-aged women with HT revealed practically the same reduction in BP after resistance training with or without green tea supplementation (Figure 3). There was no significant difference between the placebo and green tea intake groups (500 mg daily = 245 mg polyphenol, 75 mg epigallocatechin gallate, 25 mg caffeine) or placebo (490 mg maltodextrin). The participants performed circuit training consisting of two sets with a resistance of 50% one repetition maximum (1RM). BP was measured at zero, 15, 30, 45, and 60 min after training.

Wong et al. [78] observed the BP changes in postmenopausal obese women with a BMI ≥ 25 kg/m^2^. There was a similar decrease in the brachial and arterial BP in all groups (*p* < 0.05). Vibration training in combination with the placebo, vibration training with L-citrulline or L-citrulline alone can be used as effective ways to lower BP (Figure 3). However, there was no improvement in body composition between the study groups.

Romero et al. [71] observed the effects of folic acid supplementation for six weeks among seniors. The authors found that the folic acid intake immediately before training reduces HR but not mean arterial pressure. At the end of the six-week experiment, the HR values and mean arterial pressure were higher than the baseline values (*p* = 0.05 compared to acute folic acid ingestion). A significant positive feature is the fact that folic acid increases the blood flow to active skeletal muscles, mainly due to better local vasodilation.

## 4. Discussion

The present study is the first systematic review to analyze the evidence for the effectiveness of ST combined with the NP or supplementation on the BP values. This systematic research analyzed the results of 303 individuals from nine studies. Studies by Moraes et al. [70], Villani, Gornall [74], Sales et al. [75], Figueroa et al. [76], and Lee et al. [73] compared the effects of ST combined with a NP on BP. All of these studies used different methods and parameters for the ST protocol and the NP. A common feature of these five studies was the reductions in SBP and DBP or mean arterial pressure in all experimental groups, regardless of the type of training or diet. Only four studies with supplements met the criteria for inclusion in the systematic review: Astorino et al. [72], Arazi et al. [77], Wong et al. [78] and Romero et al. [71]. The effects of caffeine, L-citrulline, folic acid, and green tea were investigated. Only two supplements had a positive impact on BP: L-citrulline and green tea. On the other hand, caffeine and folic acid did not decrease BP, and caffeine has even been reported to increase BP.

In general, the issue of ST and diet or supplementation has been considered as completely different disciplines, which should be examined separately. However, in the case of treatment of PHT and HT, their combination may appear to be the most effective method. Although this study was not able to conclude whether ST alone or ST combined with the NP has the greatest impact on arterial pressure reduction, it successfully summarized current studies comparing different ST and NP interventions.

### 4.1. Blood Pressure Reduced by Strength Training Alone

It is generally known that ST increases strength, muscle mass, and bone mass and simultaneously helps reduce the symptoms of various chronic diseases, such as heart disease [81,82,83,84,85]. In 2005, the AHA (American Heart Association) started to recommended ST for lowering BP [2] because ST induces a post-exercise BP decrease, as supported by many reviews [14,15,16,86]. One of the first long-term ongoing studies conducted on cardiovascular disease was the Framingham Heart Study [24], where one of the findings reported was that a 2 mmHg reduction in DBP was associated with an estimated 17% decrease in the prevalence of HT. However, unsuitable training parameters such as the work load, reps per set, rest interval, etc., can increase BP [87] above the recommended values, i.e., 220/105 mmHg [1]. This study summarize that ST alone can decrease SBP from 132 ± 4 mmHg to 125 ± 2 mmHg [76] or from 141 ± 2 mmHg to 132 ± 16 mmHg [78], and can decrease DBP from 82 ± 3 mmHg to 77 ± 2 mmHg [76], which is much more than the smallest significant values of 2 mmHg. The one-time effect of ST induced a SBP decrease (from 136.88 ± 5.9 mmHg to 117.82 ± 6.09 mmHg) in the study by Arazi et al. [77] and in the prehypertensive group of the study by Astorino et al. [72] (from 143 ± 11.4 mmHg to 131.7 ± 16.6 mmHg). However, in a study by Arazi et al. [77], the biggest BP reduction was not observed immediately after training but after 1 h. There is great diversity between the training sessions in this systematic review. None of the included studies was identical in terms of the training parameters, methods, or exercises (Table 2), but all included studies set up the ST intervention according to the recommended parameters and methods published by Pescatello et al. [80]. Moreover, no study reported a dangerous increase of BP over 220/105 mmHg (according to the ACSM) or substantial post-exercise hypotension. According to the results of this review, ST alone can be recommended as an effective nonpharmacological intervention and prevention method for people with HT or PHT.

### 4.2. Blood Pressure was Reduced by the Nutrition Program

Nutrition-based approaches are recommended as a first-line therapy for the prevention of HT [72], where the AHA recommends a specific program called the DASH diet to treat and prevent HT. However, some food components such as alcohol, sodium, simple sugar, and saturated fat, have been shown to increase BP [9,51,88]. It has also been found that for HT reduction, weight loss is essential, which has been shown to reduce BP in overweight hypertensive and prehypertensive individuals [10,89,90,91,92]. This systematic review included only one study that used the DASH diet. The Korean variation of the DASH diet in a study by Lee et al. [73] improved SBP and DBP only in conjunction with ST (from 139.3 ± 13.2 mmHg to 135.7 ± 15.3 mmHg). Diet alone increased SBP (from 135.3 ± 11.8 mmHg to 135 ± 9.6 mmHg), but decreased DBP (from 86.7 ± 9.2 mmHg to 81.1 ± 8.2 mmHg [73]. Conversely, Villani, Gornall [74], Sales et al. [75] and Figueroa et al. [76] included hypocaloric diets, which led to a reduction in BP in both groups (ST group and NP alone). Moreover, it emerged that ST alone had similar effectiveness as the NP alone [76]. Based on this review result, we recommend a NP that takes into account the individual components of the food and that also leads to a drop in BP. Moraes et al. [70] showed that the inclusion of more dairy products in the NP, together with training, can have a positive effect on the lowering of the SBP (from 138.3 ± 4.6 mmHg to 135.2 ± 4.5 mmHg) and DBP (from 91.3 ± 5.3 mmHg to 88.3 ± 4.9 mmHg) values. The same higher intake of dairy products resulted in a smaller increase in BP (i.e., return of BP to the initial values) after six weeks of nontraining. It seems that a higher intake of dairy products prolongs training hypotension. Therefore, multiple effects of ST and a NP should not be expected for the BP lowering. Although their combination is not more effective, their combination might result in more health benefits (such as weight loss) than ST and NP alone.

### 4.3. Effect of Supplements on Blood Pressure

Caffeine is a central nervous stimulant whose physiological effects for increasing sport performance are extensive, but sometimes conflicting or contradicting [93,94]. Caffeine supplementation has been shown to decrease feelings of fatigue and promote mood and perceptual responses during any exercise, including ST [94,95,96]. It has been found that SBP and DBP increase after caffeine ingestion due to the vasoconstrictive effects of caffeine [93]. However, some studies have reported mixed results regarding caffeine intake and BP. In randomized controlled trials (RCTs), short-term caffeine intake caused an acute increase in SBP and DBP by 2/1 mmHg, respectively, compared with the effects of decaffeinated coffee or abstinence [8,97]. Caffeine supplementation, according to the results of Astorino et al. [72], cannot be recommended in individuals with HT or PHT because it not only increases the resting BP but also maintains the BP at a high level after the end of the workout. Therefore, the hypotensive effect of ST is completely lost [72]. For that reason, we cannot recommend caffeine to individuals with HT or PHT although caffeine reduces body fat, increases sport performance, and delays fatigue [94,95,96].

L-citrulline is a precursor of L-arginine, a substrate for nitric oxide synthase, in the production of nitric oxide. Deficiencies in the L-arginine supply have been strongly implicated in cardiovascular diseases, including HT, atherosclerosis, diabetic vascular disease, hyperhomocysteinemia, heart failure, etc. [98,99,100]. In a study by Wong et al. [78], L-citrulline decreased BP in individuals performing ST or not performing ST. Both combinations led to a reduction, but slightly better results were achieved in the group that performed the ST alone. Here, again, this study confirmed that ST plays a primary role in the entire process of hypotension. The main and indisputable advantage of L-citrulline is that it reduces BP alone but also in combination with NP or ST. However, more studies are needed to explore this supplement.

A link between green tea and BP reduction has been explored for decades in Chinese populations [101]. However, there are few studies regarding the long-term effects of tea drinking on the risk of HT, and the results of the few studies investigating the relationship between tea consumption and BP were opposing. In epidemiological studies, a higher consumption of black tea in Norwegian individuals was associated with a lower SBP [102], while the green tea intake in Japanese self-defense officials was unrelated to BP [103]. Long-term effects of green tea consumption do not reduce the values of SBP or DBP after training in comparison with the effects of training alone. In contrast, a significant difference between the two groups was observed in the increase in HR after the end of the training. The green tea group performing RT showed a smaller increase in HR than the training group alone. These positive effects can be attributed, at least in part, to the antioxidant properties and vasodilating effects of the catechins in green tea [104]. Unlike caffeine, green tea is not dangerous for individuals with a high BP because it does not increase BP. For this reason, it can be included for its stimulating effects as a preworkout drink. A very significant drop in the systolic blood pressure was measured in the study by Arazi et al. [77], where blood pressure dropped from preintervention values from 133.12 mmHg (± 3.7) to 116.25 mmHg (± 3.71) in the group that combine ST and green tea. According to this information, green tea can be recommended as a suitable food supplement for the treatment of HT. However, supplementation with green tea together with ST does not lead to a bigger reduction in BP in comparison with the effects of the ST alone.

The primary risk factor for a stroke, is HT as stated by Meschia et al. [105]. For this reason, the effect of folic acid on the frequency of infarcts was investigated. Most of the relevant randomized trials were designed for secondary prevention and have not shown a beneficial effect of folic acid for cardiovascular disease prevention [106,107,108,109]. Based on this study result, folic acid had an effect on the HR and mean arterial pressure [71]. The acute folate intake before exercise induced a reduction in HR but not a long-term decrease. Even the long-term intake did not lead to reduction. For all measurements, a similar increase in HR and mean arterial pressure during isometric exercises was observed with the acute and long-term intake of folic acid [71]. Based on this study, folic acid supplementation in elderly individuals does not lead to a significant decrease in the mean arterial BP or HR during exercise compared to the measurements of the control subjects [71]. Similar results were observed in studies by Huo et al. [63] and Hernández-Díaz et al. [110]. According to these findings, we cannot recommend folic acid alone for lowering BP.

### 4.4. Combined Effect of Strength Training, Nutrition Program and Supplementation on Blood Pressure

Although BP lowering is effective using ST alone [76,77,78], NP alone [74,76] and some food supplements alone [77,78], the combination of ST, NP, and supplementation did not have additional systolic BP lowering effectivity (Appendix A). On the other hand, the combination of ST and NP seems to be necessary to decrease the diastolic BP [70,78] (Appendix A). Moreover, the combination of ST with green tea or L-citrulline decreased the systolic BP on a higher significance or effect size (Appendix A) than the ST alone, although the significant difference between the ST and ST with supplementation has not been directly reported. This findings supports the notion that total behavioral modification such as lifestyle modifications, rather than focusing on modifying a single behavioral target, is more important. Therefore, a combination of ST with the energy intake restriction NP, which include food supplement stimulants green tea and L-citrulline should be used for BP lowering and additional health benefits such as body composition changes.

### 4.5. Changes in Body Composition

Body composition improvements have been shown as a result of ST and NPs in studies by Villani et al. [74], Sales et al. [75], Figueroa et al. [76], Moraes et al. [70], and Lee et al. [73]. Slightly better results, in terms of improved body composition, were achieved by a NP alone in the studies by Villani et al. [74] and Figueroa et al. [76]. On the other hand, better results were not achieved by the DASH alone compared with the DASH diet and a training program in the latest study by Lee et al. [73]. Only a small improvement in the body composition was recorded by Figueroa et al. [76] in the ST group. Furthermore, Figueroa et al. [76] found that a hypocaloric diet decreased the brachial–ankle pulse-wave velocity mainly by reducing the leg pulse-wave velocity, and this reduction was related to fat loss. Although ST alone does not affect the pulse-wave velocity or body composition, ST combined with NP improves the brachial–ankle pulse-wave velocity and muscle strength while preventing the loss of lean body mass in obese postmenopausal women. The results of the pulse-wave velocity are considered an independent predictor of systolic hypertension [111].

In this case, a NP appears to be important for body composition transformation. However, the advantages of ST cannot be overlooked, and their combination seems to be the optimal variant. This systematic review reported some important findings concerning the different effects of various training, nutrition, and supplementary strategies on HT. Individuals with a high BP can improve their BP values, not only by using medication but also by utilizing a nonpharmacological method, which will not only have a positive impact on BP but also on other health components, such as body composition, muscle strength, and bone density. Future studies should more closely examine the effect of specific training, nutrition, and supplementation programs on individuals with HT, normotension, and PHT. Additionally, future studies should explore mechanisms of how different nutrition programs and supplements lower BP.

## 5. Conclusions

Presented data suggest that a combination of ST and NP seems to be an effective solution for lowering BP (systolic and diastolic). This method can be recommended for individuals with PHT or HT. However, an essential role can be attributed to the strength training program alone. In both aerobic and anaerobic forms, exercise causes post-exercise hypotension, but the effect depends on the selected training parameters. A NP based on the restriction of energy intake positively affects BP and body fat. However, this study cannot recommend caffeine supplementation because it increases BP during ST, while a more suitable stimulant is green tea or L-citrulline, which lower BP. For BP lowering in clinical practice, it is recommended to prescribe a combination of ST with the energy intake restriction NP, which include food supplements stimulants green tea and L-citrulline.

## Figures and Tables

**Figure 1 jcm-08-01393-f001:**
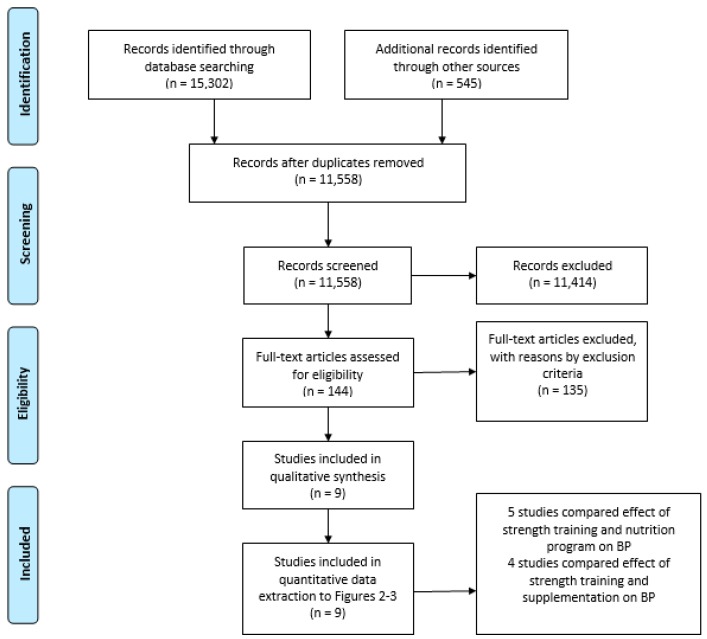
Flowchart of the selection process. BP = blood pressure.

**Figure 2 jcm-08-01393-f002:**
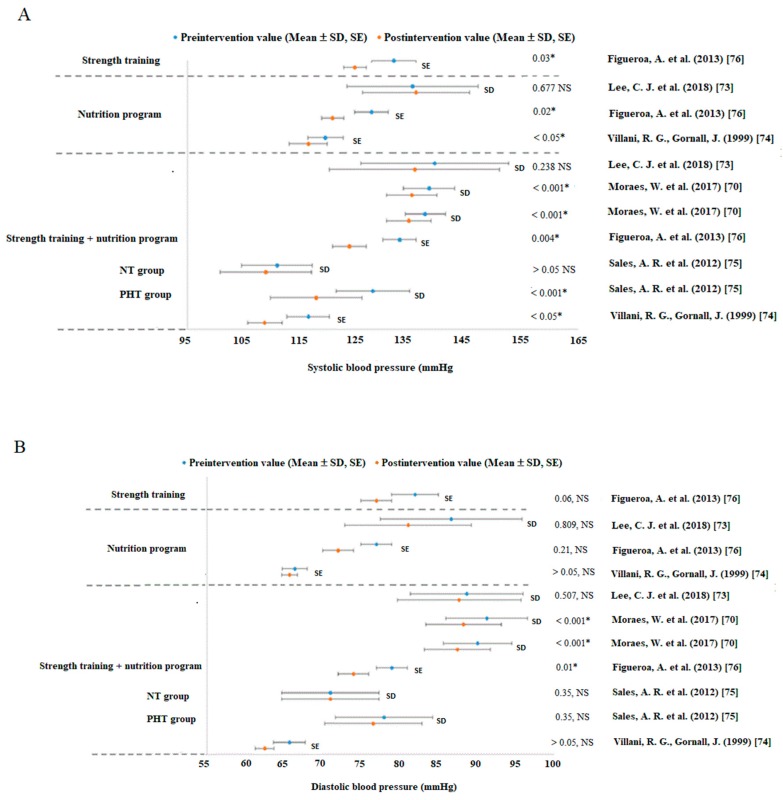
Changes in systolic (**A**) and diastolic (**B**) blood pressure (mmHg) in nutrition and strength training program studies. Abbreviations: NP = nutrition program, NT = normotension group, PHT = prehypertension group, SD = standard deviation, SE = standard error, ST = strength training, NS = not significant change—no change; * significant difference on reported *p* value.

**Figure 3 jcm-08-01393-f003:**
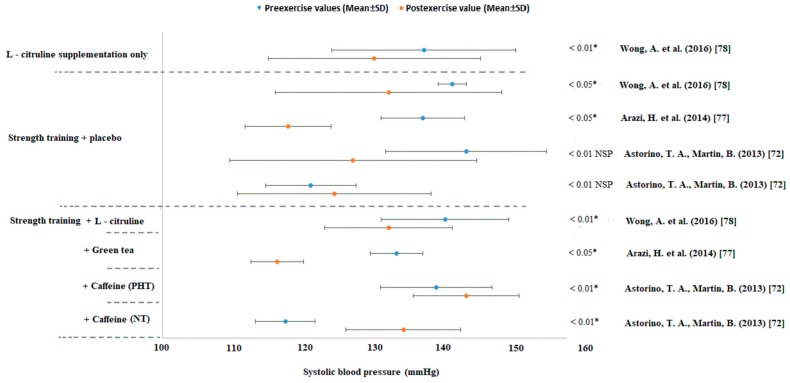
Changes in systolic blood pressure (mmHg) in supplement studies. Abbreviations: SD = standard deviation, * significant difference on reported “*p*” value. NSP = not significant according to the post hoc test.

**Table 1 jcm-08-01393-t001:** General description of each study, sample, and intervention characteristics.

Authors	Subjects	Aim	Results
Villani and Gornall (1999) [74]	Premenopausal women NP-only group: n = 10, mean age (y) = 33, weight (kg) = 75.78 ± 3.2 NP + Strength training group: n = 10, mean age (y) = 33, weight (kg) = 79.50 ± 2.86	The aim of the study was to investigate the combined influences of very-low-kilojoule diets and strength training on BP.	Resistance exercise did not significantly alter the BP reduction observed with short-term severe dieting.
Sales et al. (2012) [75]	Women with prehypertension: n = 10, age (y) = 39 ± 6, weight (kg) = 71.5 ± 10.7 Women with normotension n = 10, age (y) = 35 ± 11, weight (kg) = 66.5 ± 6.8	The aim of the study was to investigate the effect of diet and exercise training on BP and autonomic modulation in women with prehypertension.	Diet and exercise training reduced SBP in women with prehypertension, and this was associated with parasympathetic augmentation and probably reduction in sympathetic cardiac modulation.
Astorino and Martin (2013) [72]	Hypertensive men: n = 7, age (y) = 23.9 ± 4.6, height (m) = 1.8 ± 0.1, mass (kg) = 89 ± 16.2 Normotensive men: n = 7, age (y) = 22.4 ± 4.0, height (m) = 1.8 ± 0.1, weight (kg) = 77.9 ± 6.4	The primary aim of the study was to compare changes in BP in normotensive and prehypertensive men completing resistance exercise following caffeine ingestion.	Post-exercise hypotension did not occur in either treatment, suggesting that intense resistance training with or without caffeine intake may mitigate the BP-lowering effect of resistance exercise.
Figueroa et al. (2013) [76]	Postmenopausal women LIRET: n = 14, age (y) = 54 ± 1, height (m) = 1.66 ± 0.02, weight (kg) = 88.4 ± 4.6 Postmenopausal women NP: n = 13, age (y) = 54 ± 1, height (m) = 1.62 ± 0.02, weight (kg) = 89.0 ± 4.4 Postmenopausal women NP + LIRET: n = 14, age (y) = 54 ± 1, height (m) = 1.63 ± 0.02, weight (kg) = 86.7 ± 2.7	The aim of the study was to evaluate the independent and combined effects of a hypocaloric diet and LIRET with slow movement on PWV and body composition.	A hypocaloric diet decreases baPWV mainly by reducing legPWV, and this reduction was related to the loss of truncal fat. Although LIRET alone does not affect PWV or body composition, LIRET combined with diet improves baPWV and muscle strength while preventing loss of lean body mass in obese postmenopausal women.
Arazi et al. (2014) [77]	Middle-aged women: n = 24, age (y) = 46.4 ± 6.3, height (m) = 1.66 ± 4.2, weight (kg) = 66.6 ± 9.2 kg	The aim of the study was to investigate the effects of green tea extract on BP, HR, and RPP responses to low-intensity resistance exercise in hypertensive women.	Three weeks of green tea extract ingestion did not influence SBP, DBP or HR but may be have a favorable effect on MAP and RPP responses to an acute resistance exercise during a 1-h exercise recovery.
Wong et al. (2016) [78]	Postmenopausal women whole-body vibration training + Placebo: n = 14, age (y) = 58 ± 4.0, height (m) = 1.6 ± 0.06, weight (kg) = 89.5 ± 10.6 Postmenopausal women L-citrulline: n = 14, age (y) = 58 ± 4.0, height (m) = 1.6 ± 0.05, weight (kg) = 83.8 ± 8.4 Postmenopausal women WBVT + L-citrulline: n = 13, age (y) = 58 ± 3.0, height (m) = 1.62 ± 0.05, weight (kg) = 88.3 ± 3.9	The aim of the study was to examine the combined and independent effects of whole-body vibration training and L-citrulline supplementation on aortic hemodynamics and plasma nitric oxide metabolites in postmenopausal women.	This study supports the effectiveness of whole-body vibration training + L-citrulline as a potential intervention for the prevention of hypertension-related cardiac diseases in obese postmenopausal women.
Moraes et al. (2017) [70]	Low milk intake group: n = 16, age (y) = 70.2 ± 4.9 and weight (kg) = 70.1 ± 7.6 High milk intake group: n = 12, age (y) = 70.3 ± 5.0, weight (kg) = 68.6 ± 7.7	The aim of the study was to investigate whether the maintenance of exercise training benefits are associated with adequate milk and dairy product intake in elderly hypertensive subjects after detraining.	Maintenance of exercise training benefits related to pressure levels, lower extremity strength and aerobic capacity is associated with adequate milk and dairy product intake in hypertensive elderly subjects following six weeks of detraining.
Romero et al. (2017) [71]	Adults (men and women): n = 9, age (y) = 68 ± 5, height (m) = 1.65 ± 5, weight (kg) = 70 ± 8	The purpose of this study was to test the hypothesis that folic acid ingestion improves skeletal muscle blood flow in aged adults performing graded handgrip and plantar flexion exercise via increased vascular conductance.	Folic acid ingestion increases blood flow to active skeletal muscle primarily via improved local vasodilation in aged adults.
Lee et al. (2018) [73]	Adults (men and women) Advice-only comparison group: n = 28, age (y) = 43.4 ± 14.5, weight (kg) = 69.9 ± 9.2 Diet education group: n = 30, age (y) = 43.0 ± 13.5, weight (kg) = 72.8 ± 15.2 Diet and exercise education group: n = 27, age (y) 49.1 ± 10.1, weight (kg) = 76.6 ± 10.7	The aim of this study was to evaluate the effectiveness of a home-based lifestyle modification intervention on BP management.	Lifestyle modification emphasizing both diet and exercise was effective for lowering BP and should be favored over diet-only modifications.

baPWV, brachial–ankle pulse-wave velocity; BP, blood pressure; DBP, diastolic blood pressure; HR, heart rate; LIRET, low-intensity resistance exercise training, PWV, pulse-wave velocity; MAP, mean arterial pressure; RPP, rate pressure product; SBP, systolic blood pressure; y, year.

**Table 2 jcm-08-01393-t002:** Strength training parameters and methods in each study.

Authors	Sets	Rest Between Sets	Repetitions	Intensity	Frequency	Strength Training Methods	Number of Exercises
Villani and Gornall (1999) [74]	3	1–2 min	10	10 RM	3x per week	Resistance training	6
Sales et al. (2012) [75]	3	-	10	45–85% 1RM	3x per week	Resistance + aerobic training	10
Astorino and Martin (2013) [72]	4	2 min	As much as possible	70–80% 1RM	2 measurements	Resistance free weight training	4
Figueroa et al. (2013) [76]	2–3	-	18–22x	-	3x per week	Resistance machine training	4
Arazi et al. (2014) [77]	2	2 min	6–10	50% 1 RM	1 measurement	Resistance training	6
Wong et al. (2016) [78]	1–5	0.5–1 min	0.5–1 min	25–40 Hz	3x per week	Bodyweight training	8
Moraes et al. (2017) [70]	-	-	-	-	2x per week	Resistance + endurance + flexibility + stability training	-
Romero et al. (2017) [71]	2	20 min	1	3, 6, 9 kg	2 measurements	Isometric training	2
Lee et al. (2018) [73]	3	-	-	Mild–moderate intensity	Every day	Circuit training	5–7

RM = repetition maximum.

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
