# Peer review of "Role of Nutrition and Exercise Programs in Reducing Blood Pressure: A Systematic Review"

_jcm, 2019, doi:10.3390/jcm8091393_

Round 1

Reviewer 1 Report

Page 1, paragraph 2, lines 3 & 5 - suggest amend such that sentence does not start with an abbreviation

Page 2, paragraph 2, lines 1 & 5 - suggest amend to avoid first person language

2.1, line 1 - remove track change (and) & amend punctuation (...Scopus, and...')

Page 3, paragraph 1, line 2 - suggest amend to '...task, or the...

3.1, paragraph 3, line 1 - suggest amend to avoid first person language '...compilation, it can be stated...

Page 4, Figure 1 - refers to a meta-analysis, but no reporting of such is evident

Page 7, 3.2, paragraph 2, line 9 - suggest amend such that sentence does not start with an abbreviation

lines 10, 15 & 16 - suggest amend to avoid first person language 

line 15 - abbreviation 'DASH' not defined prior to use here (is subsequently defined on line 101)

line 23- abbreviation 'PHT' not defined prior to use here

Page 9, line 38 - suggest amend such that sentence does not start with an abbreviation

line 43- abbreviation 'EGCG' not defined prior to use here

line 59 - suggest amend to avoid first person language '...best of authors' knowledge...'

page 10, multiple instances - suggest amend to avoid first person language

line 96 - 'great nonpharmacological' needs to be tempered - the authors offer no clinical significance to their Discussion or Conclusion, when in fact the changes reported are only in the magnitude of 2 to 3 mmHg

page 11, lines 115, 137 & 159 - suggest amend to avoid first person language

line 156 - suggest amend such that sentence does not start with an abbreviation

Page 12, lines 166, 192 & 196 - suggest amend to avoid first person language

line 177 - suggest amend punctuation to '...composition, ST combined...

Reference list - review for consistency with style guide - journal title abbreviating (some are, some are not), capitalisation of article title

There are 111 citations/references - this is a very large amount for what is essentially a less than 6 page manuscript - are these all essential for your message?

Tables S1-S3 - what does 'After p' mean? presumably this should be after program? - suggest be explicit for clarity. Further, it is not clear what the p-value ('baseline values' & 'after p') and effect size (baseline values) represent - what is being compared at baseline to give a p-value & effect size?

Table S4 - correct typographical error in title (Physioterapy). Further, suggest add column giving PEDro score for the articles (rather than making the reader calculate from all other columns)

Author Response

Thank you for your valuable time spend on our manuscript, we improved our manuscript according to your suggestions. 

Reviewer 1

Page 1, paragraph 2, lines 3 & 5 - suggest amend such that sentence does not start with an abbreviation

Answer: Done we amended this part of text and check similar flows in all other text.

Page 2, paragraph 2, lines 1 & 5 - suggest amend to avoid first person language

Answer: Done we overwrite the text without first person.

2.1, line 1 - remove track change (and) & amend punctuation (...Scopus, and...')

Answer: Done

Page 3, paragraph 1, line 2 - suggest amend to '...task, or the...

Answer: Done

3.1, paragraph 3, line 1 - suggest amend to avoid first person language '...compilation, it can be stated...

Answer: Done, according to suggestion.

Page 4, Figure 1 - refers to a meta-analysis, but no reporting of such is evident

Answer: Thank you for this point, the original aim to perform meta-analyses was not possible due to the data type, which were not enough consistent. Instead mentioning meta-analyses we have added type of outcome comparison an amend it the Figure 1 to be more specific. In supplementary material we also calculated effect size by Cohen d, but this was just for better orientation and not main result.

Page 7, 3.2, paragraph 2, line 9 - suggest amend such that sentence does not start with an abbreviation

Answer: Done, according to suggestion.

lines 10, 15 & 16 - suggest amend to avoid first person language 

Answer: Done.

line 15 - abbreviation 'DASH' not defined prior to use here (is subsequently defined on line 101)

Answer: This abbreviation has been defined at this point.

line 23- abbreviation 'PHT' not defined prior to use here

Answer: This abbreviation has been defined at this point.

Page 9, line 38 - suggest amend such that sentence does not start with an abbreviation.

Answer: Done.

line 43- abbreviation 'EGCG' not defined prior to use here

Answer: We have written “epigallocatechin gallate” without abbreviation, since there is no more abbreviation further in the text.

line 59 - suggest amend to avoid first person language '...best of authors' knowledge...'

Answer: Done.

page 10, multiple instances - suggest amend to avoid first person language

Answer: Done.

line 96 - 'great nonpharmacological' needs to be tempered - the authors offer no clinical significance to their Discussion or Conclusion, when in fact the changes reported are only in the magnitude of 2 to 3 mmHg.

Answer: We agree that this stamen is too strong, if based on presented numerical values. Therefore, we have added in this paragraph the examples of Blood pressure improvement/changes which were more that 3mmHg. However, we used the term “effective” instead “great”  in discussion and conclusion.

page 11, lines 115, 137 & 159 - suggest amend to avoid first person language

Answer: Done

line 156 - suggest amend such that sentence does not start with an abbreviation

Answer: Done

Page 12, lines 166, 192 & 196 - suggest amend to avoid first person language

Answer: Done

line 177 - suggest amend punctuation to '...composition, ST combined...

Answer: Done

Reference list - review for consistency with style guide - journal title abbreviating (some are, some are not), capitalisation of article title.

Answer: We carefully check the journal guideline and put all journals with official abbreviation abbreviated. Some unnecessary Capitalisation in article title has been corrected. Thus, now the references are due to the journal guidelines.

There are 111 citations/references - this is a very large amount for what is essentially a less than 6 page manuscript - are these all essential for your message?

Answer: We believe that all of our references are appropriate and necessary, since this review is quite interdisciplinary combining strength training, nutrition (including supplements) and blood pressure. All of those issues are quite large areas, which deserve references for many of their points. Our reference list includes a large number of other reviews, meta-analyses and articles cover by respected associations, which we used with aim to use rather lower amount of references (it would be really too much to used more of original studies). With respect to keep text flow together with evidence based arguments, we ended up with this reference number.

Tables S1-S3 - what does 'After p' mean? presumably this should be after program? - suggest be explicit for clarity. Further, it is not clear what the p-value ('baseline values' & 'after p') and effect size (baseline values) represent - what is being compared at baseline to give a p-value & effect size?

Answer: Sorry for this inaccuracy, we accidentally uploaded older version of supplementary material. Now we revised the table into correct form, where we clearly reporting Pre and Post intervention “p” values (with type of intervention) and “p” values between different intervention reported from same studies. The effect sizes were additionally calculated by Cohen d since most studies did not provided effect sizes directly in their reporting.

Table S4 - correct typographical error in title (Physioterapy). Further, suggest add column giving PEDro score for the articles (rather than making the reader calculate from all other columns).

Answer: Thank you for this comment, we have added the column with total PEDro scores and its verbal interpretation. We also corrected typographical error.

Reviewer 2 Report

Your review is focused on nutrition and exercise training affected high BP improvement. In your review you compare systematic search of strength training (ST) program alone, nutrition program (NP) alone and their combinations from 9 studies from1999 to 2018 year with 111 article.

Your data suggest the best lowering BP in combination of ST and NP and in NP stimulants green tea and L-citrulline.

Comments:

Abstract:

Change 1966 to 1999

Results:

In Figures 2-4:Mean PRE, mean Post- include whole words Correct significance level in Fig 2,3,4

Fig 2, 3: Some outputs does  not look as significance level: Villani 1999, Moraeis 2017, Sales 2012

Fig 4:  In Arazi 2014 is missing significance level output, which should be around 0.01, while Astori (2013) strength training and caffeine, does not look significant.

In Figures 2 and 3 there are  Changes in systolic and diastolic blood pressure (mmHg) and in nutrition program studies.

So, it should be written as Fig 2 using A , B  of systolic and diastolic changes: Changes in systolic (A) and diastolic (B) blood pressure      (mmHg) in nutrition program studies

Last Fig 4 is about Blood pressure changes in supplement studies. It should be change to Fig 3

Dissussion:

In 4.1. and 4.2 Blood pressure reduced by strength training  and by the nutrition program. Include more BP info in text about  mmHg and/or % changes. Include more info (e.g to new 4.5. part) about lowering of BP in combination of ST and NP

Author Response

Your review is focused on nutrition and exercise training affected high BP improvement. In your review you compare systematic search of strength training (ST) program alone, nutrition program (NP) alone and their combinations from 9 studies from 1999 to 2018 year with 111 article.

Your data suggest the best lowering BP in combination of ST and NP and in NP stimulants green tea and L-citrulline.

Answer: Thank you for your time to revise our manuscript and effective conclusion of main results. We have corrected the article according to your suggestions, and also included the triple combination of our findings (ST, NP + green tea and L-citrulline) as the specific conclusion for clinical practice.

Comments: Abstract:

Change 1966 to 1999

Answer: Done

Results:

In Figures 2-4:Mean PRE, mean Post- include whole words Correct significance level in Fig 2,3,4

Answer: We have corrected, added or put more specific description of significance levels in our Figures. We have added the specification, whether the reported significance value was significant or not significant in analyses performed in reported study. Another specification was also showing whether SD or SE was reported. We want to show also some non-significant results. Now the figures are much more informative. Along with that we also improved the clarity of figures.

Fig 2, 3: Some outputs does  not look as significance level: Villani 1999, Moraeis 2017, Sales 2012

Answer: Sorry for this inaccuracy, some studies used SD and some SD. We double check all reported values and corrected them if relevant (some SD were wrong). We added directly whether the authors analyses reported significant or nonsignificant differences.

Fig 4:  In Arazi 2014 is missing significance level output, which should be around 0.01, while Astori (2013) strength training and caffeine, does not look significant.

Answer: We doublecheck all reported values and corrected them if relevant. Arazi 2014 significance level has been added, however this author decided to use for his analyses p values 0.05 (the values would be probably significant also on 0.01, but we can change the original authors approach). In Astorino 2013 study, we marked which reported values has significance and which does not, where some values were not different according to post hoc test. This happened because Astorino 2013 used 2x11x2 or 2x 4 repeated measure ANOVA, where differences were reported based on post hoc test.

In Figures 2 and 3 there are  Changes in systolic and diastolic blood pressure (mmHg) and in nutrition program studies. So, it should be written as Fig 2 using A , B  of systolic and diastolic changes: Changes in systolic (A) and diastolic (B) blood pressure (mmHg) in nutrition program studies.

Answer: Thank you for this point, we now renamed the figures as 2 A and 2 B and renumbered Figure 4 to Fig 3.

Last Fig 4 is about Blood pressure changes in supplement studies. It should be change to Fig 3

Answer: Done

Dissussion:

In 4.1. and 4.2 Blood pressure reduced by strength training  and by the nutrition program. Include more BP info in text about  mmHG and/or % changes.

Answer: Thank you for this suggestion we have added more values right into the text.

Include more info (e.g. to new 4.5. part) about lowering of BP in combination of ST and NP.

Answer: We have added the “4.4 Combined effect of strength training, nutrition program and supplementation on blood pressure “, which summarize the program combinations and actually answer our hypothesis. This was actually missing in the first version.

Reviewer 3 Report

Roman Jurik et al. performed systematic review and meta-analysis for the publications of non-pharmacological blood pressure lowering intervention from 1966 to 2018. Their analyses provide important insight into the fields of blood pressure management in which currently pharmacological interventions are dominantly adopted. Their findings about equal contribution of both nutrition program and strength training are interesting and recommendation of combination of ST 2-3 times a week at moderate intensity and NP is reasonable. 

Author Response

Thank you for your positive evaluation and spending time on our manuscript. We did our best to make this manuscript clear and beneficial for further studies.

Round 2

Reviewer 2 Report

Dear authors,

you improved and finalised your article.

Article should be accepted.